# Survival after Lung Metastasectomy from Esophageal Cancer: Results from a Multi-Institutional Database

**DOI:** 10.3390/cancers15051472

**Published:** 2023-02-25

**Authors:** Yoshikane Yamauchi, Jun Nakajima, Mingyon Mun, Yasushi Shintani, Hiroaki Kuroda, Takekazu Iwata, Makoto Endo, Yoko Azuma, Masayuki Chida, Yukinori Sakao, Ichiro Yoshino, Norihiko Ikeda, Haruhisa Matsuguma, Kazuhito Funai, Hiroshi Hashimoto, Masafumi Kawamura

**Affiliations:** 1Department of Surgery, Teikyo University School of Medicine, Tokyo 170-8582, Japan; 2Department of Thoracic Surgery, Graduate School of Medicine, The University of Tokyo, Tokyo 113-8655, Japan; 3Department of Thoracic Surgical Oncology, Cancer Institute Hospital, Tokyo 135-8550, Japan; 4Department of General Thoracic Surgery, Osaka University Graduate School of Medicine, Suita 565-0871, Japan; 5Department of Thoracic Surgery, Aichi Cancer Center Hospital, Nagoya 464-8681, Japan; 6Department of Thoracic Surgery, Chiba Cancer Center, Chiba 260-8670, Japan; 7Department of Thoracic Surgery, Yamagata Prefectural Central Hospital, Yamagata 990-2292, Japan; 8Division of Chest Surgery, Department of Surgery, School of Medicine, Toho University, Tokyo 143-8541, Japan; 9Department of General Thoracic Surgery, Dokkyo Medical University, Mibu 321-0293, Japan; 10Department of General Thoracic Surgery, Graduate School of Medicine, Chiba University, Chiba 260-8670, Japan; 11Department of Thoracic Surgery, Tokyo Medical University, Tokyo 160-0023, Japan; 12Division of Thoracic Surgery, Tochigi Cancer Center, Utsunomiya 320-0834, Japan; 13First Department of Surgery, Hamamatsu University School of Medicine, Hamamatsu 431-3192, Japan; 14Department of Thoracic Surgery, National Defense Medical College, Saitama 359-8513, Japan

**Keywords:** lung metastasis, esophageal cancer, pulmonary metastasectomy, prognostic factor

## Abstract

**Simple Summary:**

To clarify the clinical impact and to identify prognostic predictors of surgical intervention for pulmonary metastasis from esophageal cancer, a registry database analysis was performed. A database was developed by the Metastatic Lung Tumor Study Group of Japan from January 2000 to March 2020. One hundred and nine cases were reviewed and examined the prognostic factors for pulmonary metastasectomy of metastases from esophageal cancer. As a result, five-year overall survival after pulmonary metastasectomy was 34.4%, and five-year disease-free survival was 22.1%. The multivariate analysis for overall survival and that for disease free survival revealed significant prognostic factors, such as initial recurrence site, maximum tumor size, duration from primary tumor treatment to lung surgery, number of lung metastases, and preoperative chemotherapy for lung metastasis. In conclusion, eligible patients with pulmonary metastasis from esophageal cancer selected based on the identified prognostic predictors would be good candidates for pulmonary metastasectomy.

**Abstract:**

To clarify the clinical impact and to identify prognostic predictors of surgical intervention for pulmonary metastasis from esophageal cancer, a registry database analysis was performed. From January 2000 to March 2020, patients who underwent resection of pulmonary metastases from primary esophageal cancer at 18 institutions were registered in a database developed by the Metastatic Lung Tumor Study Group of Japan. An amount of 109 cases were reviewed and examined for the prognostic factors for pulmonary metastasectomy of metastases from esophageal cancer. As a result, five-year overall survival after pulmonary metastasectomy was 34.4% and five-year disease-free survival was 22.1%. The multivariate analysis for overall survival revealed that initial recurrence site, maximum tumor size, and duration from primary tumor treatment to lung surgery were selected as the significant prognostic factors (*p* = 0.043, *p* = 0.048, and *p* = 0.037, respectively). In addition, from the results of the multivariate analysis for disease free survival, number of lung metastases, initial recurrence site, duration from primary tumor treatment to lung surgery, and preoperative chemotherapy for lung metastasis were selected as the significant prognostic factors (*p* = 0.037, *p* = 0.008, *p* = 0.010, and *p* = 0.020, respectively). In conclusion, eligible patients with pulmonary metastasis from esophageal cancer selected based on the identified prognostic predictors would be good candidates for pulmonary metastasectomy.

## 1. Introduction

Esophageal cancer is one of the most aggressive of all gastrointestinal malignancies. It is the eleventh most common cause of cancer worldwide (473,000 cases) and the sixth most common cause of cancer-related mortality (436,000 deaths) [1]. Five-year relative survival for all stages of esophageal cancer was only 20% in the United States [2], although advances in multimodal treatment have recently been achieved [3]. In particular, no effective or beneficial systemic treatment has been established for esophageal carcinoma that has metastasized to distant sites. The anticipated incidence of clinically detected distant metastases from esophageal carcinoma ranges from 27.3% to 66.7% [4,5]. Among the most common sites of metastasis is the lungs [5,6]. However, the prognosis is differs depending on the site of metastasis, and it is reported that the five-year overall survival (OS) rates were better in patients who underwent lung surgical resection than in those who underwent surgical resection of other organs [6].

In general, pulmonary metastasectomy is one of the standard methods of treatment for patients with pulmonary metastases [7,8,9]. In fact, the resection of metastatic lung tumors derived from a variety of malignancies, including colorectal, uterine, head and neck, urinary tract, hepatocellular, and gastric cancers, has become the standard of care in highly selective cases [10,11,12,13,14]. However, for patients with esophageal cancer, the clinical factors that best predict prognosis remain unknown, although several studies conducted in small numbers of patients have already examined the role of pulmonary resection [5,15,16,17]. In this study, a retrospective analysis was performed using a registry database of the Metastatic Lung Tumor Study Group of Japan (METAL-J) to clarify the clinical impact and to identify prognostic predictors of surgical intervention for pulmonary metastasis from esophageal cancer.

## 2. Materials and Methods

### 2.1. Database

The METAL-J developed a database to register cases of lung metastases since 1984, and a total of 7251 cases have already been registered. All these patients underwent surgical resection for metastatic lung tumors. This study included cases in which surgery was performed for the purpose of treatment of metastatic lung tumors. Therefore, lung resections for biopsy were excluded. The database contains the following parameters: sex, age, primary tumor status, treatment for primary tumor, metastatic tumor status, date and details of metastases resected, disease-free survival (DFS), OS, and follow-up. This METAL-J registry study was approved by the institutional ethics committee of Teikyo University (current version of approval number: 19-013, approved on 12 April 2019) and other institutions.

In this registry, clinical and pathological data from patients with pulmonary metastases from esophageal cancer were collected at 18 institutions in Japan. From these data, patients who underwent curative lung metastasectomy from January 2000 to March 2020 were included in this study. The exclusion criteria were as follows: (1) Cases with unknown prognosis and cases with unknown details of lung surgery; (2) patients with residual tumor in the thoracic cavity on the operative side; and (3) patients with follow-up less than 90 days after lung surgery. Figure 1 shows the flowchart of data selection. This retrospective study was also approved by the institutional ethics committee of Teikyo University (approval number: 21-142, approved on 4 November 2021).

Radiological diagnosis, indications for surgery, and pathological diagnosis of metastatic lung tumors were all dependent on the respective institutions. At each participating institution, all the decisions related to the diagnosis and treatment of the tumors were made by an institutional cancer board consisting of oncologists, radiologists, pathologists, surgeons, and related specialists. In general, the metastasectomy was intended to achieve macroscopically complete removal of all pulmonary lesions in patients without any radiographical evidence of extrathoracic metastasis or sign of uncontrolled primary tumors. Regarding the presence of lymph node metastasis, lymph node dissection or sampling was performed in patients who underwent lobectomy or segmentectomy, but not in patients who underwent wedge resection. Wedge resection is performed only when there is no evidence of lymph node metastasis on preoperative PET-CT or other methods.

### 2.2. Statistical Analysis

SPSS version 28 (IBM Corporation, Armonk, NY, USA) and GraphPad Prism version 9.4 (GraphPad Prism Software Inc., San Diego, CA, USA) were used for statistical analyses and to construct figures. A *p*-value of <0.05 was considered statistically significant. The optimal cutoff values for continuous prognostic indexes were determined with the method established by Budczies et al. [18] described at https://molpathoheidelberg.shinyapps.io/CutoffFinder_v1/ (accessed on 25 February 2023). This method fits Cox proportional hazard models to the dichotomized variable and the survival variable. The optimal cutoff was defined as the point with the most significant split. OS was defined as the time between pulmonary metastasectomy and death or last follow-up date. DFS was defined as the time between pulmonary metastasectomy and further recurrence, death, or last follow-up date. Patients alive at the date of the last follow-up were censored. Survival curves according to the clinicopathological factors were depicted by means of the Kaplan–Meier method, and comparisons between the curves were performed by the log-rank test. Cox proportional models were used for univariate and multivariable analyses to assess the relationships between the clinicopathological factors and survival after pulmonary metastasectomy. In the Cox proportional analysis, the continuous variables were analyzed instead of using the aforementioned optimal cutoff values.

## 3. Results

A total of 109 patients with pulmonary metastasis from esophageal carcinoma were eligible for the analysis. Table 1 shows the clinicopathological characteristics of the patients. Ninety-seven (89%) were male and one hundred and three (94.5%) of primary tumors were squamous cell carcinoma. As an initial treatment for primary tumor, 36 patients (33%) received esophagostomy alone, 28 patients (26%) received surgery and chemotherapy, 21 patients (19%) received chemotherapy and radiotherapy, and 12 patients (11%) received surgery, chemotherapy, and radiotherapy. Fifteen cases (14%) had detected and treated metastases to other sites prior to the discovery of lung metastases. Multiple lung nodules were detected in 24 cases at the discovery of lung metastases. Fourteen cases (13%) received neoadjuvant chemotherapy for lung metastasis and sixteen cases (15%) received adjuvant chemotherapy, while two patients (2%) received both neoadjuvant and adjuvant chemotherapy. Eighty-six cases (79%) underwent wedge resection for lung metastasis. Three patients (2.8%) had lymph node metastases identified at surgery that were accessible from the ipsilateral thoracic cavity. The mean maximum tumor diameter of resected lung metastases was 16.9 ± 7.7 mm. The mean interval between the first primary treatment and lung surgery was 887 ± 684 days. Disease-free interval from the initial treatment was 664 ± 527 days. After lung metastasis surgery, the median follow-up period was 20 months, and further recurrence was observed in 62 patients, while 54 patients eventually died.

Figure 2 shows the survival curve after surgery for lung metastasis. Three-year OS was 48.3% and five-year OS was 34.4%. Median OS was 33.8 months. However, three-year DFS was 40.3% and five-year DFS was 22.1%, while median DFS was 26.6 months.

Figure 3 shows a comparison of OS curves regarding patients’ characteristics using the log-rank test. When the maximum tumor size of lung metastasis was more than 18 mm, the OS of patients was significantly worse than that of patients with lung metastasis of not less than 18 mm (*p* = 0.027; Figure 3a). Furthermore, when multiple lung metastases were found at the time of lung metastasis detection, OS of patients was significantly worse than that of patients with solitary lung metastasis at the time of lung metastasis detection (*p* = 0.012; Figure 3b). When the surgery for lung metastasis was performed less than 800 days after the initial treatment of the primary tumor, the OS of patients was significantly worse than that of patients with surgery performed not less than 800 days after the initial treatment of their primary tumor (*p* = 0.002, Figure 3c). Finally, when the lymph node metastases identified at surgery were accessible from the ipsilateral thoracic cavity, OS was significantly worse than that without lymph node metastasis (*p* = 0.003, Figure 3d).

Figure 4 shows a comparison of DFS curves regarding co-factors among patients’ characteristics. When the maximum tumor size of lung metastasis was more than 18 mm, DFS of patients was significantly worse than that of patients with lung metastasis of not less than 18 mm (*p* = 0.047; Figure 4a). In addition, when multiple lung metastases were found at the time of lung metastasis detection, DFS was significantly worse than that with solitary lung metastasis at lung metastasis detection (*p* = 0.003; Figure 4b). When the surgery for lung metastasis was performed at less than 720 days after the initial treatment of the primary tumor, DFS was significantly worse than that with the surgery performed not less than 720 days after the initial treatment of the primary tumor (*p* = 0.013; Figure 4c). Finally, when the lymph node metastases found at surgery were accessible from the ipsilateral thoracic cavity, DFS was significantly worse than that without lymph node metastasis (*p* = 0.02; Figure 4d).

Table 2 shows the result of Cox regression multivariate analysis for OS and DFS after lung metastasectomy. 

On the basis of the above log-rank analyses, as well as co-factors found to be significant in the previous literature, the following items were selected for the multivariate analysis: number of lung metastases at detection, lymph node metastasis found at lung metastasectomy, initial recurrence site, maximum tumor size of resected lung metastasis, duration from primary tumor treatment to lung surgery, and preoperative chemotherapy for lung metastasis. In the analysis of OS, initial recurrence site, maximum tumor size, and duration from primary tumor treatment to lung surgery were selected as the significant prognostic factors (*p* = 0.043, *p* = 0.048, and *p* = 0.037, respectively). However, in the analysis of DFS, number of lung metastases, initial recurrence site, duration from primary tumor treatment to lung surgery, and preoperative chemotherapy for lung metastasis were selected as the significant prognostic factors (*p* = 0.037, *p* = 0.008, *p* = 0.010, and *p* = 0.020, respectively). On the other hand, lymph node metastasis and distant metastasis of primary tumors and their treatment did not affect the prognosis after lung metastasis surgery (Appendix A).

## 4. Discussion

This study indicated that long-term survival can be expected in selected patients after pulmonary metastasectomy for esophageal cancer. Furthermore, univariate and multivariate analyses identified the following factors as being associated with good prognosis: single lung metastasis, initial metastasis in the lung, tumor diameter ≤18 mm, duration between primary tumor treatment and lung surgery >2 years, and administration of preoperative chemotherapy for lung metastasis. Although pulmonary metastasectomy has been reported as an effective treatment strategy that can be expected to provide long-term survival in various solid tumors, few studies reported the results of pulmonary metastasectomy for esophageal cancer, perhaps because of the extremely poor prognosis of esophageal cancer. This study included 109 cases of pulmonary metastasectomy, which is the largest number of cases compared with any of the previous reports. Thus, the results obtained in this study may provide reliable prognostic factors regarding the impact of pulmonary metastasectomy.

Prognosis after recurrence of esophageal cancer is very poor, and the five-year survival rate after recurrence is reported to be approximately 5% [19]. However, it was demonstrated that the survival rate of patients with fewer recurrence sites was better than that of patients with multiple recurrence sites [20]. In addition, curative treatment of oligo-recurrence will improve survival, and many papers reported that oligo-recurrence may be a favorable characteristic of tumor biology [19,20,21,22,23,24]. In this respect, pulmonary metastasectomy should be an important treatment option for selected patients with lung metastasis. Moreover, Depypere et al. showed prolonged survival in patients with solitary local recurrence or single solid organ metastases, especially when surgery was performed [19]. In particular, Nobel et al. reported that patients with lung oligometastasis had significantly longer median OS than patients with other sites of oligometastasis [23]. They reported that the median OS of patients with lung oligometastasis was 2.41 years, which is quite similar to our study.

Here, the question may arise as to who a good candidate for pulmonary metastasectomy is. The lung is one of the major recurrence sites from esophageal cancer [25], but metastasectomy will contribute to a better prognosis only in very selected cases. In this selection, the prognostic predictors identified in our study would be helpful for clinicians, and it is clear that the tumor number and size should be defined. The results of the prognostic predictors in the OS analysis suggest that, in cases of preceding recurrence in other organs or recurrence within 800 days from the initial treatment, curative surgery would likely be indicated, preferably after a minimum 800 days, or 2.2 years, after the first treatment, provided that chemotherapy is administered before surgery to prevent further metastatic lesions. On the other hand, neither disease-free interval after the initial treatment nor the time from the initial treatment to the discovery of lung metastases were the significant factors in the univariate and multivariate analysis. This suggests that the impact of pulmonary metastasectomy depends on the timing of the surgery, because the time from the initial treatment for the primary tumor until lung metastasectomy was important, but not so much the disease-free interval after the initial treatment or the time from the initial treatment for the primary tumor until the discovery of lung metastasis. The prognosis of lung tumor treatment was not significantly associated with the N or M factor of the primary tumor or the content of the initial treatment. This database only includes cases in which lung resection was performed for the treatment of metastatic lung tumors. Therefore, we believe that information that might affect the prognosis of the primary tumor was irrelevant to the prognosis after lung resection because of that very selective selection of cases from all cases of esophageal cancer.

Conversely, some would argue that the prognosis after pulmonary metastasectomy should account for the possibility that primary lung squamous cell carcinoma might be mixed in the population of “lung metastasis from esophageal cancer”, considering that prognosis improved with a longer duration from the initial treatment for the primary tumor. The risk factors for carcinogenesis associated with esophageal squamous cell carcinoma and lung squamous cell carcinoma are very similar [26,27]. In fact, some cases may be difficult to classify pathologically, even if immunohistochemical staining is performed. In a study of squamous cell carcinomas arising in the lungs after treatment for squamous cell carcinoma derived from the head and neck region, Gurts et al. verified the concordance of clinical information, pathological evaluation, and genetic information by loss of heterozygosity analysis, and determined that half of the cases with a clinical diagnosis of metastatic lung tumor were likely to be primary lung cancers [28]. In this registry study, patients were enrolled who were determined to have lung metastases by the tumor board at each participating institution, and there is no information on the presence or absence of genetic proof of metastasis. Therefore, the possibility of primary lung cancer being mixed in the study population cannot be ruled out. However, it has been reported that a similar trend was observed in a group of patients with complete resection of esophageal cancer, including other recurrence sites [22], suggesting that the inclusion of primary lung cancer is not enough to account for this trend. Furthermore, five-year OS after wedge resection for solid-type squamous cell carcinoma of the lung (<2 cm) is reported to be approximately 55% [29]; this prognosis is considerably better than that of the current study group, suggesting that there may be a small number of lung cancer cases in this study population, if any. It is true that it is difficult to distinguish between metastatic and primary lung cancer based solely on a partial tissue biopsy, in view of tumor heterogeneity. In addition, even if the patient has primary lung cancer, surgery can be expected to contribute to the prognosis, even if it is sublobar resection [30]. Therefore, we believe that a curative surgical approach should be justified in cases that meet the prognostic predictors shown in this study.

This study has several limitations. First, it was a retrospective study using a registry database whose information was prospectively collected. Therefore, measurement bias was inevitable to a certain extent. We should consider the possibility that potential confounding factors have not been taken into account because of the insufficient information in the database, such as whether nonoperative treatment methods were used as preoperative or postoperative adjuvant therapy and whether the surgical method was open, thoracoscopic, or robotic. Second, the significance of surgery is not fully evaluated because this study was conducted on patients who underwent surgical treatment and did not compare the results with those of patients who did not undergo surgical treatment. However, lung resection is superior to other treatment methods in that it allows pathological diagnosis and treatment of the lesion at the same time. As mentioned above, it is difficult to distinguish esophageal cancer pulmonary metastases from primary lung cancer pathologically, so sufficient pathology specimens would be always required. From this point of view, we believe that lung resection should be recommended in suitable cases. Furthermore, this is a multi-institutional study, and each institution had autonomy over who should be enrolled in this study; therefore, selection bias was inevitable to a certain extent. Moreover, because the study period covered 20 years, not only the preoperative diagnostic procedures but also the treatment may not be uniform due to advances made in the clinic during this time, such as PET-CT and chemotherapy including immunotherapy. It is assumed that not all cases included in this study were diagnosed and treated equally in this regard, and the possibility that the results of this study strongly reflect the influence of some treatments cannot be ruled out. However, even with these limitations, it would be useful to analyze the impact of pulmonary metastasectomy in a large population of patients and to identify prognostic predictors for future treatment selection.

## 5. Conclusions

The surgical indication of pulmonary metastasectomy derived from esophageal cancer should be determined based on prognostic predictors such as tumor size, tumor number, initial metastasis site, duration between primary tumor treatment and metastasectomy, and administration of preoperative chemotherapy for pulmonary metastasis. Highly selected patients with pulmonary metastasis from esophageal cancer would be good candidates for pulmonary metastasectomy, although some primary lung cancer cases can be included in the population.

## Figures and Tables

**Figure 1 cancers-15-01472-f001:**
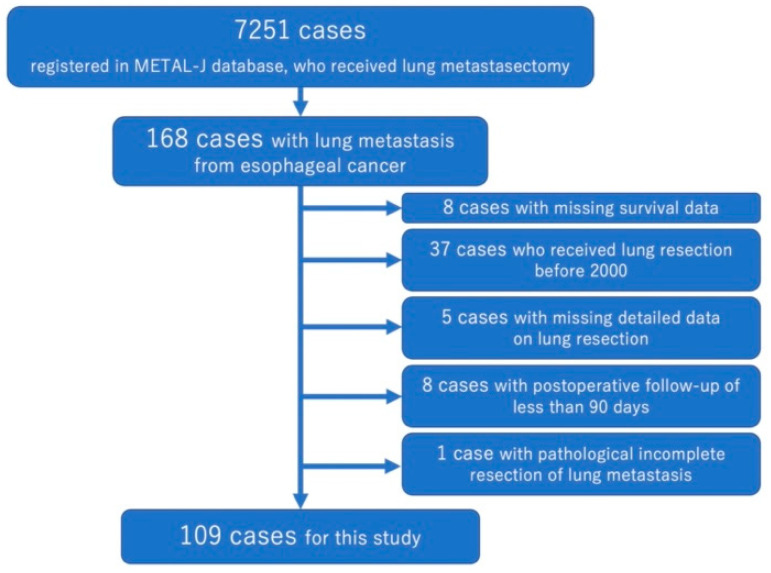
The flowchart of patient selection.

**Figure 2 cancers-15-01472-f002:**
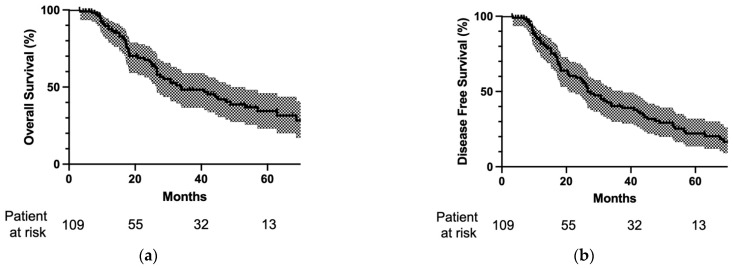
(**a**) Overall survival and (**b**) disease-free survival after pulmonary metastasectomy. The survival curves are depicted as solid lines and 95% confidence intervals are drawn in halftone. The numbers of patients at risk at lung surgery at 20, 40, and 60 months after lung surgery are reported at the bottom of the curves. Three-year overall survival was 48.3% and five-year overall survival was 34.4%. Three-year disease-free survival was 40.3% and five-year disease-free survival was 22.1%.

**Figure 3 cancers-15-01472-f003:**
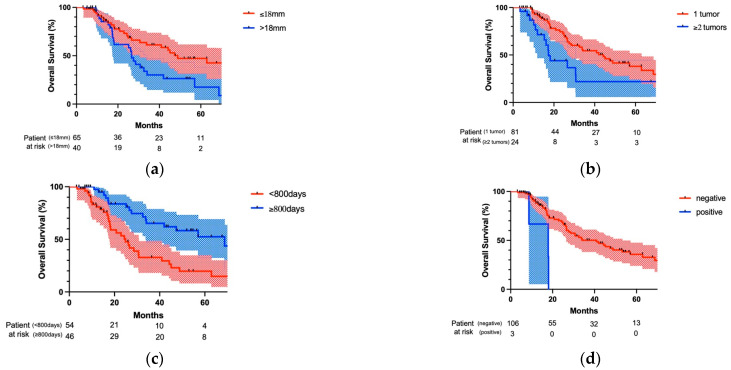
Comparison of overall survival after pulmonary metastasectomy regarding patients’ characteristics. (**a**) Maximum tumor size of lung metastasis was more than 18 mm and OS was significantly worse than that of patients with lung metastasis of not less than 18 mm (*p* = 0.027). (**b**) When multiple lung metastases were found at the time of lung metastasis detection, OS was significantly worse than that with solitary lung metastasis at the lung metastasis detection (*p* = 0.012). (**c**) When the surgery for lung metastasis was performed less than 800 days after the initial treatment of the primary tumor, OS was significantly worse than that when the surgery was performed not less than 800 days after the initial treatment of the primary tumor (*p* = 0.002). (**d**) When the lymph node metastases found at surgery were accessible from the ipsilateral thoracic cavity, OS was significantly worse than that without lymph node metastasis (*p* = 0.003).

**Figure 4 cancers-15-01472-f004:**
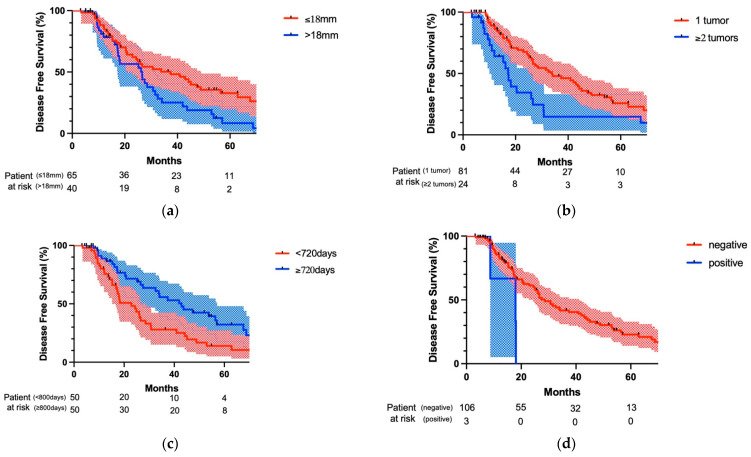
Comparison of disease survival after pulmonary metastasectomy regarding patients’ characteristics. (**a**) When maximum tumor size of lung metastasis was more than 18 mm, DFS was significantly worse than that with lung metastasis of not less than 18 mm (*p* = 0.047). (**b**) When multiple lung metastases were found at the time of lung metastasis detection, DFS was significantly worse than that with solitary lung metastasis at lung metastasis detection (*p* = 0.003). (**c**) When the surgery for lung metastasis was performed less than 720 days after the initial treatment of primary tumor, DFS was significantly worse than that when the surgery was performed not less than 720 days after the initial treatment of the primary tumor (*p* = 0.013). (**d**) When the lymph node metastases found at surgery were accessible from the ipsilateral thoracic cavity, DFS was significantly worse than that without lymph node metastasis (*p* = 0.02).

**Table 1 cancers-15-01472-t001:** Patients’ characteristics.

Variables		Number (%)/Average
Sex		
	Male	97 (89.0)
	Female	12 (11.0)
Histology of primary tumor		
	SqCC	103 (94.5)
	ADC	3 (2.8)
	Others	3 (2.8)
Initial treatment for primary tumor		
	Surgery alone	36 (33.0)
	Chemotherapy alone	3 (2.8)
	Radiotherapy alone	1 (0.9)
	Surgery and Chemotherapy	28 (25.7)
	Surgery and Radiotherapy	3 (2.8)
	Chemotherapy and Radiotherapy	21 (19.3)
	Surgery, Chemotherapy, and Radiotherapy	12 (11.0)
	Not available	5 (4.6)
pathological N stage of primary tumor		
	N0	27 (24.8)
	N1–3	63 (57.8)
pathological M stage of primary tumor		
	M0	73 (67.0)
	M1	19 (17.4)
Initial recurrence site		
	Pulmonary	94 (86.2)
	Extrapulmonary	15 (13.8)
Number of tumors at detection of lung metastasis		
	1	81 (74.3)
	2–4	24 (22.0)
perioperative chemotherapy for lung metastasis		
	Before lung surgery	14 (12.8)
	After Lung surgery	16 (14.7)
The type of surgery for lung metastasis		
	wedge resection	86 (78.9)
	segmentectomy	9 (8.3)
	lobectomy	14 (12.8)
The side of surgery for lung metastasis		
	right	59 (54.1)
	left	38 (34.9)
	both side	12 (11.0)
Lymph node metastasis found at lung surgery		
	None	106 (97.2)
	Hilar lymph node metastasis	1 (0.9)
	Mediastinal lymph node metastasis	2 (1.8)
Patients’ age at lung surgery		67 ± 9
Maximum tumor size of resected lung metastasis (mm)		16.9 ± 7.7
Disease free interval from the initial treatment (days)		664 ± 527
Duration from primary tumor treatment to lung surgery (days)		887 ± 684

**Table 2 cancers-15-01472-t002:** Results of multivariate analysis for disease-free survival and overall survival after lung metastasectomy.

Factors		Overall Survival		Disease Free Survival	
		HR (95% CI)	*p*-Value	HR (95% CI)	*p*-Value
Number of lung metastases at the detection		1.69 (0.95–3.00)	0.072	1.66 (1.03–2.66)	0.037 *
Lymph node metastasis found at lung metastasectomy					
	negative	Reference	0.24	Reference	0.503
	positive	2.33 (0.58–9.42)		1.57 (0.42–5.92)	
Initial recurrence site					
	pulmonary	Reference	0.043 *	Reference	0.008 *
	extrapulmonary	2.65 (1.03–6.80)		2.89 (1.32–6.36)	
Maximum tumor size of resected lung metastasis		1.52 (1.00–2.29)	0.048 *	1.30 (0.92–1.84)	0.14
Duration from primary tumor treatment to lung surgery		0.99 (0.99–1.00)	0.037 *	0.99 (0.99–1.00)	0.010 *
Preoperative chemotherapy for lung metastasis					
	Undone	Reference	0.082	Reference	0.02 *
	Done	0.38 (0.13–1.13)		0.33 (0.13–0.84)	

HR: Hazard Ratio, CI: Confidence Interval. * *p* < 0.05

## Data Availability

Data available on request due to restrictions due to privacy and ethical reasons.

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
