# Peer review of "Survival after Lung Metastasectomy from Esophageal Cancer: Results from a Multi-Institutional Database"

_cancers, 2023, doi:10.3390/cancers15051472_

Round 1

Reviewer 1 Report

Comments to the author

I am grateful for the opportunity to review this interesting manuscript entitled: "Selection criteria for lung metastasis resection from esophageal cancer". This report is interesting because it focused on surgical resection for lung metastasis from esophageal cancer. However, your manuscript has several major problems as following.

1. I think that chemotherapy and/or radiotherapy (n=69) as an initial treatment includes both definitive CRT and neoadjuvant therapy. However, you should classify your patients into three groups at least; surgery alone, definitive CRT and neoadjuvant therapy followed by surgery. Initial treatment is very important in considering about recurrence pattern and treatment for recurrence.

2. The long-term outcomes after recurrence should be related to initial treatment and initial stage. Why didn’t you evaluate these factors in multivariate analysis?

Minor comments

Page 3, Line 128-129

You had better show the number of patients received each initial treatment.

Table 2

You suggested that duration from primary tumor treatment to lung surgery was one of prognostic predictors. In multivariate analysis, HR of it was low (0.99). How do you explain about these results?  

Author Response

Thank you very much for your valuable comments.

  1. This study included cases in which surgery was performed for the purpose of treatment of metastatic lung tumors. Therefore, lung resections for biopsy were excluded. Therefore, it is presumed that the primary tumor has been completely treated. Indeed, there are various treatment details for the primary tumor; as suggested in No. 3, we have made a list with respect to the initial treatment details. In this list, it should be categorized whether it is neoadjuvant or adjuvant therapy. However, in many cases, sufficient information has not been collected in this multicenter, retrospective studies. Therefore, it is difficult to accurately categorize them. This point has been added to the limitation.
  2. Our study focuses on information about long-term prognosis from the treatment of lung tumors. Following your suggestion, we compared the prognosis of the treatment of lung tumors with the N factor or M factor of initial tumors and the treatment details. No significant difference was found in any factors. I have summarized them in the supplement table. This database contains only cases in which lung resection was performed for the purpose of treatment of metastatic lung tumors. Therefore, we believe that this is a very selective group of cases compared to the total number of esophageal cancer cases. Therefore, we do not believe that any information that is likely to be relevant to the prognosis of the main lesion will be affected by the prognosis after lung resection. We have added a discussion on this point in the Discussion section.
  3. The content of the initial treatment was added in Table 1 and corrected result part accordingly.
  4. Since the Hazard Ratio was calculated as “day” from initial treatment to lung surgery, the impact of “one day” is rather small. Therefore, HR was calculated as nearly 1, but it was definitely smaller than 1, and the p-value was significantly small.

Reviewer 2 Report

Dear authors, thank you for submitting the interessing paper. 

I just have a few questions:

Is the surgical aproach ( minimal invasiv of Open surgery) be analyzed?

How are the lymph nodes be identfied? Frozen section? PET-CT/ Scan or is a systematic lymph node dissection be done?

How many metastasis are maximum for surgery? Pre-operative and intraoperative?

The different pre-operative diagnostic procedures during the long time of 20 years should be critically discussed and also the different medical and oncological therapeutic apoaches.

Author Response

Thank you very much for your valuable comments.

  1. The surgical approach was not analyzed. We added this point as a limitation.
  2. Regarding the presence of lymph node metastasis, lymph node dissection or sampling was performed in patients who underwent lobectomy or segmentectomy, but not in patients who underwent wedge resection. Wedge resection is performed only when there is no evidence of lymph node metastasis on preoperative PET-CT or other methods. We added this point in the Method.
  3. There was no limit for the number of tumor resection in the database, but four tumor was maximum in this study. These tumors were pathologically proved as metastases. We added the number in the table 1. This study included cases in which surgery was performed for the purpose of treatment of metastatic lung tumors. Lung resections for biopsy were excluded. In this respect, the number of tumors would be limited as the surgeon would expect complete resection of tumors preoperatively.
  4. We agree your opinion. We added this point as a limitation.

Reviewer 3 Report

In this study, a retrospective prognostic analysis was performed on patients who underwent resection of lung metastases from esophageal cancer. As a result, favorable prognostic factors were identified as single lung metastasis, initial lung metastasis, tumor diameter less than 18 mm, duration between primary tumor treatment and lung surgery >2 years, and preoperative chemotherapy for lung metastasis.

1. Although this study emphasizes the significance of surgery for lung metastasis of esophageal cancer, all cases were those who underwent surgery for lung metastasis. If the significance of surgery is to be emphasized, a comparison with cases in which surgical therapy was not performed is necessary.

2. It is assumed that the presence or absence of lymph node metastasis or metastasis to other organs has a strong influence on prognosis. Has an analysis been performed for the unified presence or absence of these?

Author Response

Thank you very much for your valuable comments.

  1. We agree with your opinion that a comparison with cases in which surgical therapy was not performed is necessary if the significance of surgery is to be emphasized. However, this study included cases in which surgery was performed for the purpose of treatment of metastatic lung tumors. Moreover, there were no other reports of resectable cases in whom other treatment than surgery was chosen intentionally. Therefore, it is difficult to show the comparison of the cases. On the other hand, lung resection is superior to other treatment methods in that it allows pathological diagnosis and treatment of the lesion simultaneously. As already mentioned, it is difficult to differentiate esophageal cancer pulmonary metastases from primary lung cancer, so adequate pathological specimen is required for the pathological evaluation. In this regard, we believe lung resection should be recommended in the cases suitable for lung resection. We added this point to the Discussion.
  2. Our study focuses on information about long-term prognosis from the treatment of lung tumors. Following your suggestion, we compared the prognosis of the treatment of lung tumors with the N factor or M factor of initial tumors. No significant difference was found in any factors. I have summarized them in the supplement table. This database only includes cases in which lung resection was performed for the treatment of metastatic lung tumors. Therefore, we believe that information that might affect the prognosis of the primary tumor was irrelevant to the prognosis after lung resection be-cause of that very selective selection of cases from all cases of esophageal cancer. We have added a discussion on this point in the Discussion.

Round 2

Reviewer 3 Report

Thank you for the corrections and additional data to the points I raised. My questions are now largely answered.

The title states "Selection criteria" for lung metastasis resection from esophageal cancer, but readers find it difficult to understand the selection criteria as it is stated in text only. Is it possible to show them in a flowchart or schematic figure?

Author Response

Thank you for your suggestions. I agree the title may mislead the contents of this manuscript. As the academic editor's suggestion, I changed the title as "Survival after lung metastasectomy from esophageal cancer: Results from a multi-institutional database".